

# Structure and stability of recombinant bovine odorant-binding protein: III. Peculiarities of the wild type bOBP unfolding in crowded milieu

Olga V. Stepanenko[1], Denis O. Roginskii[1], Olesya V. Stepanenko[1], Irina M. Kuznetsova[1], Vladimir N. Uversky[1,2] and Konstantin K. Turoverov[1,3]

[1] Laboratory of structural dynamics, stability and folding of proteins, Institute of Cytology, Russian Academy of Sciences, St. Petersburg, Russia
[2] Department of Molecular Medicine, University of South Florida, United States
[3] Peter the Great St. Petersburg Polytechnic University, St. Petersburg, Russia

Corresponding authors
Vladimir N. Uversky,
vuversky@health.usf.edu
Konstantin K. Turoverov,
kkt@incras.ru

## ABSTRACT

Contrary to the majority of the members of the lipocalin family, which are stable monomers with the specific OBP fold (a $\beta$-barrel consisting of a 8-stranded anti-parallel $\beta$-sheet followed by a short α-helical segment, a ninth $\beta$-strand, and a disordered C-terminal tail) and a conserved disulfide bond, bovine odorant-binding protein (bOBP) does not have such a disulfide bond and forms a domain-swapped dimer that involves crossing the α-helical region from each monomer over the $\beta$-barrel of the other monomer. Furthermore, although natural bOBP isolated from bovine tissues exists as a stable domain-swapped dimer, recombinant bOBP has decreased dimerization potential and therefore exists as a mixture of monomeric and dimeric variants. In this article, we investigated the effect model crowding agents of similar chemical nature but different molecular mass on conformational stability of the recombinant bOBP. These experiments were conducted in order to shed light on the potential influence of model crowded environment on the unfolding-refolding equilibrium. To this end, we looked at the influence of PEG-600, PEG-4000, and PEG-12000 in concentrations of 80, 150, and 300 mg/mL on the equilibrium unfolding and refolding transitions induced in the recombinant bOBP by guanidine hydrochloride. We are showing here that the effect of crowding agents on the structure and conformational stability of the recombinant bOBP depends on the size of the crowder, with the smaller crowding agents being more effective in the stabilization of the bOBP native dimeric state against the guanidine hydrochloride denaturing action. This effect of the crowding agents is concentration dependent, with the high concentrations of the agents being more effective.

## INTRODUCTION

Classical odorant binding proteins (OBPs) are intriguing members of the large lipocalin family, which, due to their ability to interact with different odorants (small hydrophobic

molecules of various nature and structure that have to travel from air to olfactory receptors in neurones through the aqueous compartment of nasal mucus (*Buck & Axel, 1991*; *Pevsner et al., 1988*; *Pevsner & Snyder, 1990*; *Snyder et al., 1989*)), play important but yet not completely understood role in olfaction (*Pelosi, 1994*). Typically, OBPs are monomeric carrier proteins characterized by a specific 3-D fold, known as a prototypic OBP-fold that represents a β-barrel composed by a 8-stranded anti-parallel β-sheet followed by a short α-helical segment, a ninth β-strand and disordered C-terminal tail (*Bianchet et al., 1996*; *Flower, North & Sansom, 2000*). The internal cavity of the OBP β-barrel is the binding site that can interact with the odorant molecules belonging to different chemical classes (*Vincent et al., 2004*).

Bovine OBP (bOBP) has a unique dimeric structure, which is different from the monomeric OBP fold found in the majority classical OBPs (see Fig. 1) (*Bianchet et al., 1996*). Each protomer in the bOBP dimer forms a β-barrel via interaction with the α-helical region of another protomer by means of the domains swapping mechanism (*Bianchet et al., 1996*; *Tegoni et al., 1996*). The domain swapping mechanism, being described for several dimeric and oligomeric proteins, is known to play important structural and functional roles (*Bennett, Schlunegger & Eisenberg, 1995*; *van der Wel, 2012*). It is believed that the domain swapping causes the increase in the interface area and thereby affects the overall protein stability (*Bennett, Choe & Eisenberg, 1994*; *Liu & Eisenberg, 2002*). In some cases it has been shown that the formation of the quaternary structure by means of domain swapping was responsible for the appearance of novel functions in corresponding protein monomers, functions, which were not originally present in the monomeric forms of those proteins (*Liu & Eisenberg, 2002*). Furthermore, early stages of the amyloid fibril formation are believed to be associated with the formation of domain-swapped oligomers (*van der Wel, 2012*).

Our previous studies revealed that there is a noticeable difference between the recombinant bOBP and a natural form of this protein isolated from tissues (*Stepanenko et al., 2014b*). Here, recombinant bOBP forms a stable native-like conformation with the decreased dimerization potential and therefore exists as a mixture of monomeric and dimeric variants (*Stepanenko et al., 2014b*). We designated this stable recombinant bOBP state in buffered solution as a "trapped" state with incorrect packing of α-helices and β-strands within the protein globule, which may interfere with the formation of the bOBP native state. This "trapped" state may be accumulated because the formation of the domain-swapped dimer by the bOBP represents a complex process that requires particular organization of the secondary and tertiary structures of the bOBP monomers. In other words, we hypothesized that the recombinant bOBP has perturbed packing of its α-helical region and some β-strands, and that these perturbations in packing of the secondary structure elements might affect the formation of native domain-swapped dimer (*Stepanenko et al., 2014b*).

Our previous analysis also revealed that the native dimeric form of the recombinant bOBP is formed under the mildly denaturing conditions (i.e., in the presence of 1.5 M guanidine hydrochloride (GdnHCl)) (*Stepanenko et al., 2014b*). This process requires noticeable reorganization of the bOBP structure and is accompanied by the formation of a
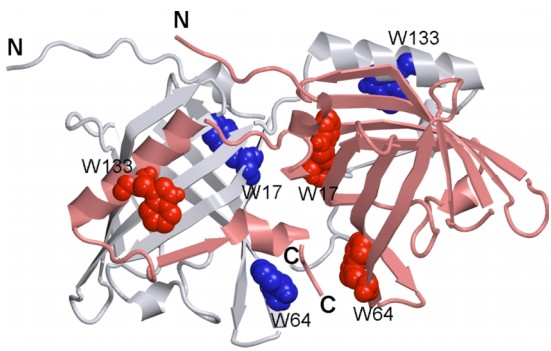

**Figure 1 3-D structure of bOBP.** The individual subunits in the protein are in gray and pink. The tryptophan residues in the different subunits are indicated in blue and red as van der Waals spheres. The drawing was generated based on the 1OBP file (*Tegoni et al., 1996*) from PDB (*Dutta et al., 2009*) using the graphic software VMD (*Hsin et al., 2008*) and Raster3D (*Merritt & Bacon, 1977*).

stable, more compact intermediate state which is maximally populated at 0.5 M GdnHCl. Cooperative unfolding of the recombinant bOBP is induced by the increase of the GdnHCl concentration above 1.5 M, whereas this protein is completed by ∼3 M GdnHCl (*Stepanenko et al., 2014b*). Thus, in the presence of GdnHCl at concentrations lower than 1.6 M, the protein molecule undergoes some local structural perturbations rather than the unfolding process. Despite its disturbed fold, the recombinant bOBP is characterized by high conformational stability, which is comparable with that of the native (isolated from tissue) bOBP (*Mazzini et al., 2002*), pOBP (*Staiano et al., 2007*; *Stepanenko et al., 2008*), and other β-rich proteins (*Stepanenko et al., 2012*; *Stepanenko et al., 2013*; *Stepanenko et al., 2014a*). This high conformational stability is indicated by the fact that the recombinant bOBP unfolding is characterized by the half-transition point of >2 M GdnHCl (*Stepanenko et al., 2014b*; *Stepanenko et al., 2016c*). We have also established that the unfolding of the recombinant bOBP is a completely reversible process, whereas the preceding process of its dimerization is the irreversible event (*Stepanenko et al., 2014b*).

One of the open challenges in the fields of protein science is the elucidation of the effects of natural cellular environment on protein structure and function, and on the processes of protein folding, unfolding, and aggregation. This challenge is defined (at least in part) by the so-called macromolecular crowding phenomenon, which originates from a known fact that the living cell contains very high concentrations of biological macromolecules (proteins, nucleic acids, polysaccharides, ribonucleoproteins, etc.), which can range from 80–400 mg/mL (*Rivas, Ferrone & Herzfeld, 2004*; *van den Berg, Ellis & Dobson, 1999*; *Zimmerman & Trach, 1991*). This crowded environment is characterized by the restricted amounts of free water (*Ellis, 2001*; *Fulton, 1982*; *Minton, 1997*; *Minton, 2000b*; *Zimmerman & Minton, 1993*; *Zimmerman & Trach, 1991*) and by the limited amount of the space available for a query protein due to the volume occupied by crowders (*Minton, 2001*; *Zimmerman & Minton, 1993*). In fact, it is estimated that the volume occupancy inside the cell is in a range of 5–40% (*Ellis & Minton, 2003*). Therefore, it is expected that in such a crowded milieu, the average spacing between macromolecules should be smaller than the size of the macromolecules themselves (*Homouz et al., 2008*),

and that the macromolecular crowding should have significant effects on various biological processes that depend on the available volume (*Minton, 2005*; *Zimmerman & Minton, 1993*).

In the laboratory practice, the potential effects of macromolecular crowding on various biological macromolecules and different biological processes are typically analyzed using solutions containing high concentrations of a model "crowding agent", such as polyethylene glycol (PEG), Dextran, Ficoll, or inert proteins (*Chebotareva, Kurganov & Livanova, 2004*; *Hatters, Minton & Howlett, 2002*; *Kuznetsova, Turoverov & Uversky, 2014*; *Kuznetsova et al., 2015*; *Minton, 2001*). Studies in this field revealed that the efficiency of crowding agents might depend on the ratio between the hydrodynamic dimensions (or occupied volumes) of the crowder and the test molecule, with the most effective conditions being those where the crowder and the test molecule occupy similar volumes (*Chen et al., 2011*; *Minton, 1993*; *Tokuriki et al., 2004*). Typically, high concentrations of inert crowders have significant effects on conformational stability and structural properties of some proteins (*Christiansen et al., 2010*; *Engel et al., 2008*; *Kuznetsova, Turoverov & Uversky, 2014*; *Mittal & Singh, 2013*), and may affect various biological processes, such as protein folding, binding of small molecules, enzymatic activity, protein-nucleic acid interactions, protein-protein interactions, protein chaperone activity, pathological protein aggregation, and extent of amyloid formation (*Chebotareva et al., 2015a*; *Chebotareva, Filippov & Kurganov, 2015b*; *Hatters, Minton & Howlett, 2002*; *Kuznetsova, Turoverov & Uversky, 2014*; *Kuznetsova et al., 2015*; *Minton, 2000a*; *Morar et al., 2001*; *Shtilerman, Ding & Lansbury, 2002*; *Uversky et al., 2002*). For example, we recently conducted a large-scale analysis of the effect of two traditional macromolecular crowders, PEG-8000 and Dextran-70, on the urea-induced unfolding of eleven globular proteins belonging to different structural classes (*Stepanenko et al., 2016a*). This analysis revealed that crowding agents do not have significant effects on the conformational stability of small, monomeric, positively charged proteins but stabilize oligomeric negatively charged proteins (*Stepanenko et al., 2016a*). Since different polymers were shown to have very different effects on the conformational stability of a given protein, it has been concluded that the excluded volume effect is not the only factor influencing the protein behavior in the crowded environments, and that the inequality of different crowders in affecting the conformational stability of proteins can be explained by the ability of the crowding agents to change the solvent properties of aqueous media (*Stepanenko et al., 2016a*).

In the first article of this series we compared structural and functional properties of the recombinant wild type bOBP and its mutants that cannot dimerize via the domain swapping (*Stepanenko et al., 2016b*). The analysis revealed that none of the amino acid substitutions introduced to the bOBP affected functional activity of the protein and that the ligand binding leads to the formation of a more compact and stable state of the recombinant bOBP and its mutant monomeric forms (*Stepanenko et al., 2016b*). Second article of the series was dedicated to the analysis of conformational stabilities of the recombinant bOBP and its monomeric variants in the absence and presence of the natural ligand (*Stepanenko et al., 2016c*). We showed that the unfolding-refolding pathways of the

recombinant bOBP and its monomeric forms are similar and do not depend on the oligomeric status of the protein, suggesting that the information on the unfolding-refolding mechanism is encoded in the structure of the bOBP monomers (*Stepanenko et al., 2016c*). Unfolding of these proteins, recombinant bOBP and its monomeric mutant forms bOBP-Gly121+ and GCC-bOBP, was accompanied by accumulation of an intermediate state that was able to bind ANS and had more compact tertiary structure than the corresponding native states. This intermediate state existed at the pre-denaturing GdnHCl concentrations, whereas the complete unfolding of these proteins proceeded from the less compact form. In the case of bOBP, the substantial unfolding of the protein precedes the subsequent transition to the native dimeric state, whereas at high GdnHCl concentrations, dissociation of this dimer occurs simultaneously with protein unfolding. Furthermore, the previous work indicated that the bOBP unfolding process is significantly complicated by the domain-swapped dimer formation, and that the rates of the unfolding-refolding reactions are controlled by the environmental conditions (*Stepanenko et al., 2016c*).

In this work, we investigated the peculiarities of the unfolding-refolding processes of the recombinant bOBP in the presence of different concentrations of model crowding agents, such as PEGs of different molecular masses. To this end, we looked at the influence of PEG-600, PEG-4000 and PEG-12000 in concentrations of 80, 150, and 300 mg/mL on the conformational stability of the recombinant bOBP against the GdnCl-induced unfolding.

## MATERIALS AND METHODS

### Materials

GdnHCl (Nacalai Tesque, Japan), ANS (ammonium salt of 8-anilinonaphtalene-1-sulfonic acid; Fluka, Switzerland) and crowding agents (PEG600, PEG4000 and PEG12000; Sigma-Aldrich, USA) were used without further purification. The protein concentration was 0.1–0.2 mg/mL. The experiments were performed in 20 mm Na-phosphate-buffered solution at pH 7.8.

### Gene expression and protein purification

The plasmid pT7-7-bOBP which encodes bOBP with a poly-histidine tag were used to transform *Escherichia coli* BL21(DE3) host (Invitrogen) (*Stepanenko et al., 2014b*). The protein expression was induced by incubating the cells with 0.3 mm of isopropyl-beta-D-1-thiogalactopyranoside (IPTG; Fluka, Switzerland) for 24 h at 37 °C. The recombinant protein was purified with Ni+-agarose packed in HisGraviTrap columns (GE Healthcare, Sweden). The protein purity was determined through SDS-PAGE in 15% polyacrylamide gel (*Laemmli, 1970*).

### Fluorescence spectroscopy

Fluorescence experiments were performed using a Cary Eclipse spectrofluorimeter (Varian, Australia) with microcells FLR (10 × 10 mm; Varian, Australia). Fluorescence intensity was corrected on the primary inner filter effect (*Fonin et al., 2014*). Fluorescence
lifetime were measured using a "home built" spectrofluorimeter with a nanosecond impulse (*Stepanenko et al., 2012*; *Stepanenko et al., 2014b*; *Turoverov et al., 1998*) as well as micro-cells (101.016-QS 5 × 5 mm; Hellma, Germany). Tryptophan fluorescence in the protein was excited at the long-wave absorption spectrum edge ($\lambda_{ex}$ = 297 nm), wherein the tyrosine residue contribution to the bulk protein fluorescence is negligible. The fluorescence spectra position and form were characterized using the parameter $A = I_{320}/I_{365}$, wherein $I_{320}$ and $I_{365}$ are the fluorescence intensities at the emission wavelengths 320 and 365 nm, respectively (*Turoverov & Kuznetsova, 2003*). The values for parameter $A$ and the fluorescence spectrum were corrected for instrument sensitivity. The tryptophan fluorescence anisotropy was calculated using the equation $r = (I_V^V - GI_H^V)/(I_V^V + 2GI_H^V)$, wherein $I_V^V$ and $I_H^V$ are the vertical and horizontal fluorescence intensity components upon excitement by vertically polarized light. $G$ is the relationship between the fluorescence intensity vertical and horizontal components upon excitement by horizontally polarized light ($G = I_V^H/I_H^H$), $\lambda_{em}$ = 365 nm (*Turoverov et al., 1998*). The fluorescence intensity for the fluorescent dye ANS was recorded at $\lambda_{em}$ = 480 nm ($\lambda_{ex}$ = 365 nm). Protein unfolding was initiated by manually mixing the protein solution (40 μl) with a buffer solution (510 μl) that included the necessary GdnHCl concentration and crowding agent concentration. The GdnHCl concentration was determined by the refraction coefficient using an Abbe refractometer (LOMO, Russia; *Pace (1986)*). The dependences of different fluorescent characteristics bOBP on GdnHCl were recorded following protein incubation in a solution with the appropriate denaturant concentration at 4 °C for different times (see in the text). The protein refolding was initiated by diluting the pre-denatured protein (in 3.0 M GdnHCl, 40 μl) with the buffer or denaturant solutions at various concentrations (510 μl), containing crowding agent. The spectrofluorimeter was equipped with a thermostat that holds the temperature constant at 23 °C.

## Circular dichroism measurements

The CD spectra were generated using a Jasco-810 spectropolarimeter (Jasco, Japan). Far-UV CD spectra were recorded in a 1 mm path length cell from 260 nm to 190 nm with a 0.1 nm step size. Near-UV CD spectra were recorded in a 10 mm path length cell from 320 nm to 250 nm with a 0.1 nm step size. For the spectra, we generated 3 scans on average. The CD spectra for the appropriate buffer solution were recorded and subtracted from the protein spectra.

## Fitting of denaturation curves

The equilibrium dependences of the parameter $A$ on the GdnHCl concentration were fit using a two-state model (*Staiano et al., 2007*):

$$S = \frac{S_N + S_U \alpha K_{N-U}}{1 + \alpha K_{N-U}}, \tag{1}$$

$$K_{N-U} = \exp\left(\frac{-\Delta G_{N-U}^0 + m_{N-U}[D]}{RT}\right), \tag{2}$$

$$K_{N-U} = F_U/F_N = (1 - F_N)/F_N, \tag{3}$$

taking into account

$$S_N = a_N + b_N[D], \tag{4}$$

$$S_U = a_U + b_U[D], \tag{5}$$

where $S$ is the parameter $A$ at the measured GdnHCl concentration; $[D]$ is the guanidine concentration; $m$ is the linear dependence of $\Delta G_{N-U}$ on the denaturant concentration; $\Delta G_{N-U}^0$ is the free energy of unfolding at 0 M denaturant; $F_N$ and $F_U$ are the fractions of native and unfolded molecules, respectively; $S_N$ and $S_U$ are the signal of the native and unfolded states, respectively; $a_N$, $b_N$, $a_U$ and $b_U$ are constants needed to fit linear dependences of the $S_N$ and $S_U$ signals on the GdnHCl concentration; and $\alpha = \frac{I_{U,365}}{I_{N,365}}$ with $I_{N,365}$ and $I_{U,365}$ being fluorescence intensity at 365 nm for the native and unfolded protein. Fitting was performed using a nonlinear regression with Sigma Plot.

Previously, to evaluate conformational stability of the studied proteins we took into account that the formation of the native dimeric state of bOBP occurred at moderate GdnHCl concentration is followed by full protein unfolding while conformational perturbations of bOBP at low denaturant concentrations were not attributed to the unfolding of the protein globule (*Stepanenko et al., 2016c*). As bOBP unfolding is fully reversible the transition from native to unfolded state of the protein was used to calculate $\Delta G_{N-U}$ value. Conformational stability of the bOBP in the crowded environment was evaluated similarly as presence of crowding agents resulted in flattering of denaturing curve of bOBP.

It is important to emphasize here that the maximal achievable concentrations of denaturant in the presence of crowding agents, especially at their highest tested concentrations, were limited by the solubility of the protein–denaturant–crowder systems. This limitation determined the number of data-points within the post-transition region.

## RESULTS AND DISCUSSION

### bOBP unfolding in the presence of PEG-600

Previously, we have shown that denaturing curves describing GdnHCl-induced unfolding of bOBP have a complex shape with two clearly distinguishable regions where the pattern of the different protein characteristics diverges significantly (*Stepanenko et al., 2014b*). In the region above 1.6 M GdnHCl, the bOBP unfolding took place as indicated by significant and simultaneous changes of all protein characteristics. The moderate structural perturbations of the bOBP with local minimum at 0.5 M GdnHCl (Figs. 2–4, red symbols and lines) in the region below 1.6 M GdnHCl were designated to the bOBP transition from a mixture of monomeric and dimeric molecules in the absence of denaturant to a native dimeric state through the local reorganization of the bOBP structure in the intermediate state at 0.5 M GdnHCl (*Stepanenko et al., 2014b*). The conformational stability of bOBP was described in terms of the half-transition values (2.1 ± 0.1 M GdnHCl, see Table 1) (*Stepanenko et al., 2016c*).

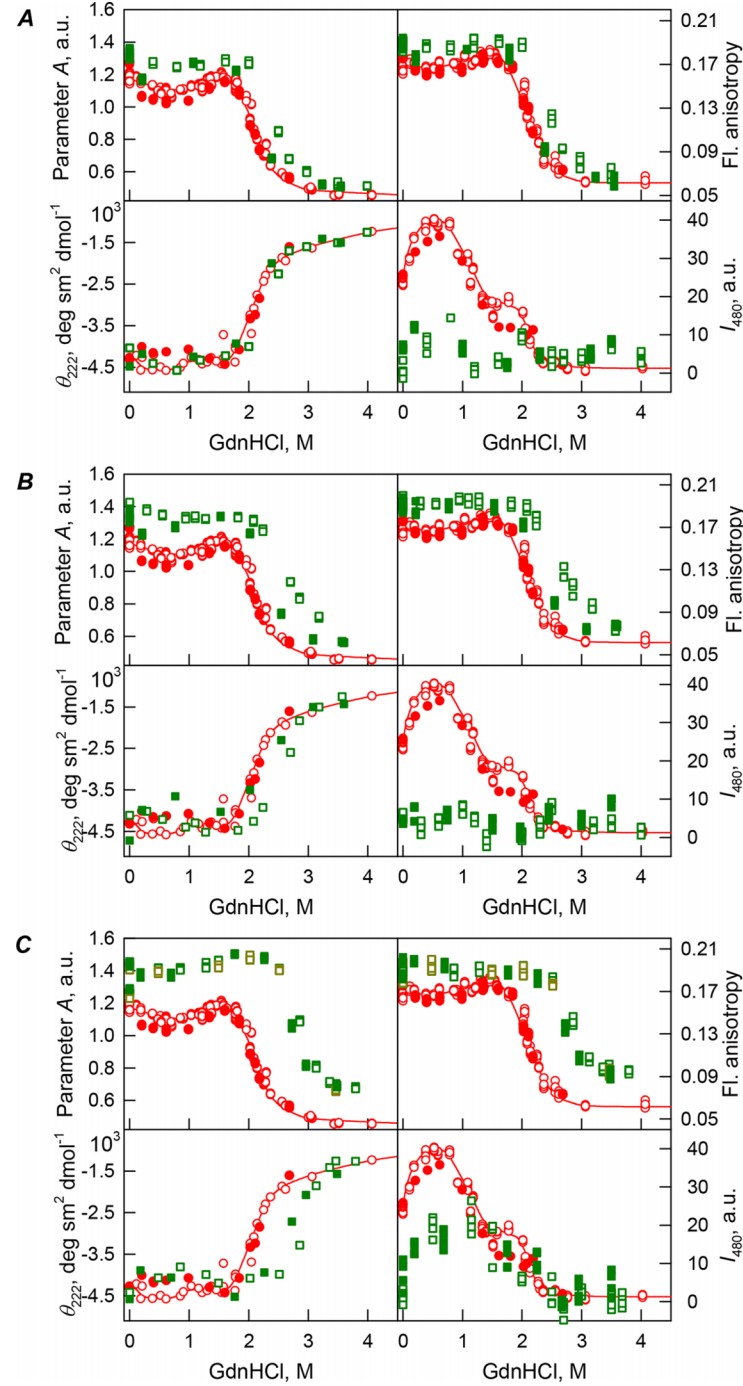

**Figure 2 GdnHCl-induced unfolding–refolding of the recombinant bOBP alone (red circles; the data are from _Stepanenko et al. (2014b)_) and in the presence of a crowding agent PEG-600 (squares) at low (80 mg/mL, (A)) medium (150 mg/mL, (B)) and high concentration (300 mg/mL, (C)).** The protein conformational changes were followed by changes in the parameter A ($\lambda_{ex}$ = 297 nm), fluorescence anisotropy $r$ at the emission wavelength 365 nm ($\lambda_{ex}$ = 297 nm), the ellipticity at 222 nm and the ANS fluorescence intensity at $\lambda_{em}$ = 480 nm ($\lambda_{ex}$ = 365 nm). Protein was incubated in a solution with the appropriate the appropriate GdnHCl concentration at 4 °C for 1 h (gray squares), 24 h (red circles), 96 h (green squares) and 7 days (dark yellow squares). The open symbols indicate unfolding, whereas the closed symbols represent refolding.

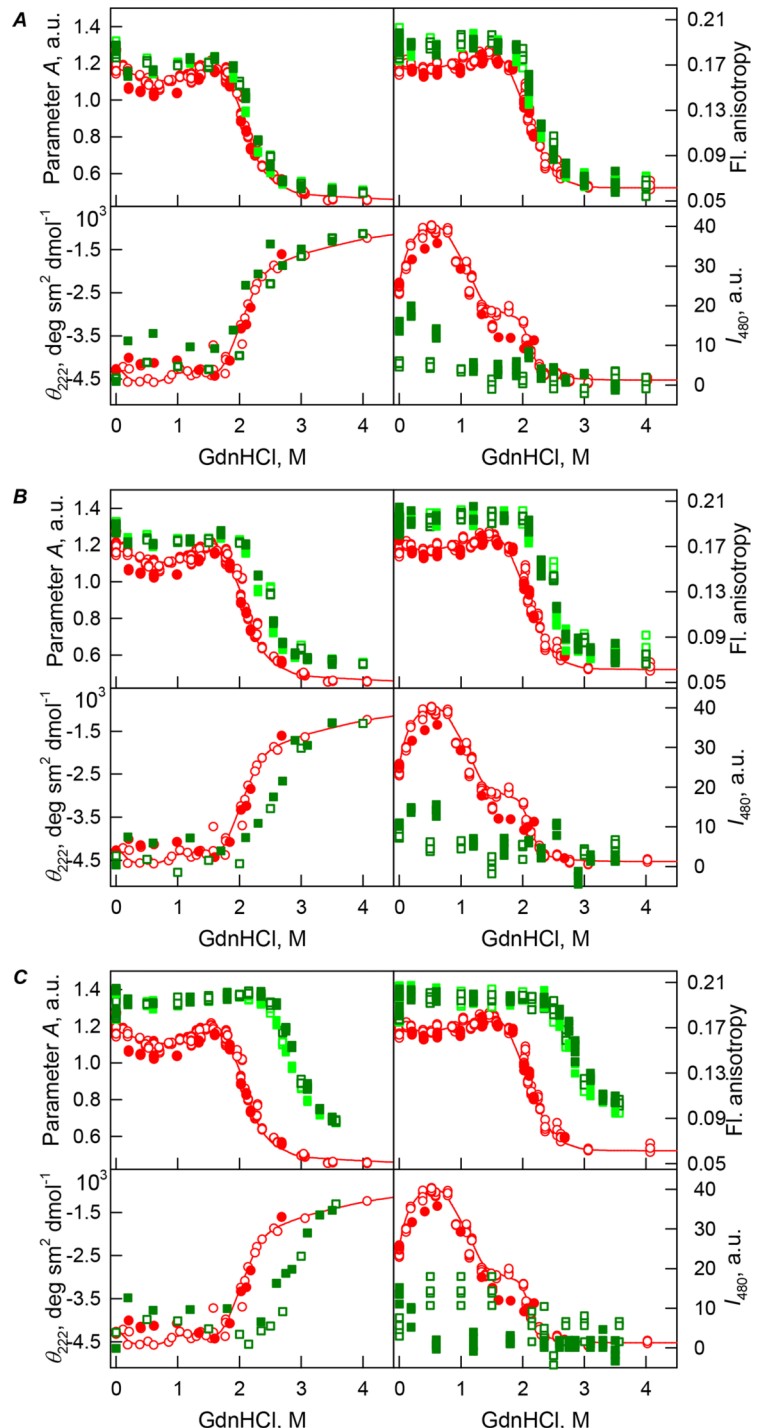

**Figure 3 GdnHCl-induced unfolding–refolding of the recombinant bOBP alone (red circles; the data are from *Stepanenko et al. (2014b)*) and in the presence of PEG-4000 (squares) at low (80 mg/L, (A)), medium (150 mg/mL, (B)) and high (300 mg/L, (C)) concentration.** The protein conformational changes were followed by the changes in the parameter $A$ ($\lambda_{ex} = 297$ nm), fluorescence anisotropy $r$ at the emission wavelength 365 nm ($\lambda_{ex} = 297$ nm), the ellipticity at 222 nm and the ANS fluorescence intensity at $\lambda_{em} = 480$ nm ($\lambda_{ex} = 365$ nm). Protein was incubated in a solution with the appropriate GdnHCl concentration at 4 °C for 1 h (gray squares), 24 h (light green squares and red circles) and 72 h (green squares). The open symbols indicate unfolding, whereas the closed symbols represent refolding.

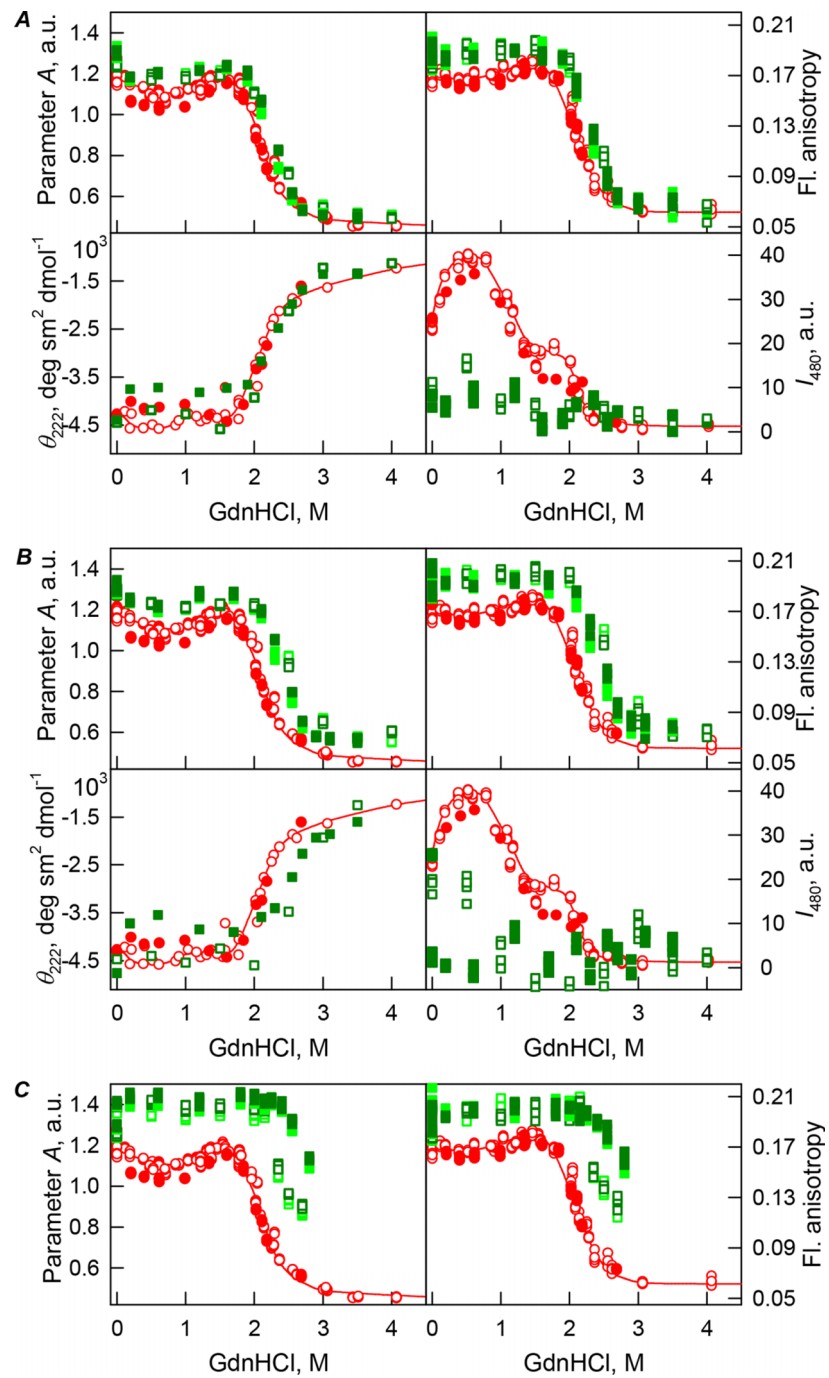

**Figure 4 GdnHCl-induced unfolding–refolding of the recombinant bOBP alone (red circles; the data are from *Stepanenko et al. (2014b)*) and in the presence of PEG-12000 (squares) at low (80 mg/mL, (A)) medium (150 mg/mL, (B)) and high concentrations (300 mg/mL, (C)).** The protein conformational changes were followed by changes in the parameter $A$ ($\lambda_{ex}$ = 297 nm), fluorescence anisotropy $r$ at the emission wavelength 365 nm ($\lambda_{ex}$ = 297 nm), the ellipticity at 222 nm, and the ANS fluorescence intensity at $\lambda_{em}$ = 480 nm ($\lambda_{ex}$ = 365 nm). Protein was incubated in a solution with the appropriate GdnHCl concentration at 4 °C for 1 h (gray squares), 24 h (light green squares and red circles) and 72 h (green squares). The open symbols indicate unfolding, whereas the closed symbols represent refolding.

**Table 1 Thermodynamic parameters of GdnHCl-induced denaturation of bOBP in the buffered solution and in the crowded environment.**

| Concentration of crowding agent | $m$ (kJ mol$^{-1}$ M$^{-1}$) | $C_m$ (M)[a] | $G_{N-U}^0$ (kJ mol$^{-1}$)[b] |
|---|---|---|---|
| Buffered solution | $3.7 \pm 0.2$ | $2.1 \pm 0.1$ | $-7.7 \pm 0.6$ |
| PEG-600 | | | |
| 80[c] | $4.0 \pm 0.4$ | $2.4 \pm 0.1$ | $-9.3 \pm 1.1$ |
| 150[c] | $2.9 \pm 0.2$ | $2.8 \pm 0.1$ | $-8.1 \pm 0.6$ |
| 300 | $3.4 \pm 0.4$ | $2.9 \pm 0.1$ | $-9.9 \pm 1.3$ |
| PEG-4000 | | | |
| 80 | $3.2 \pm 0.2$ | $2.3 \pm 0.1$ | $-7.4 \pm 0.5$ |
| 150 | $3.2 \pm 0.3$ | $2.6 \pm 0.1$ | $-8.3 \pm 0.8$ |
| PEG-12000 | | | |
| 80 | $3.1 \pm 0.2$ | $2.3 \pm 0.1$ | $-7.6 \pm 0.5$ |
| 150 | $3.5 \pm 0.5$ | $2.6 \pm 0.1$ | $-9.1 \pm 1.2$ |

Notes:
[a] $C_m$ is the denaturant concentration at midpoint of conformational transition.
[b] The fluorescence signals of the folded and unfolded states were approximated by linear dependences as function of denaturant concentration (*Nolting, 1999*).
[c] Since the unfolding curves of bOBP in the presence of 80 and 150 mg/ml of PEG-600 are quasi-equilibrium, the conformational stability of bOBP under these conditions was evaluated only for a purpose of comparison.

In other words, the formation of the native dimeric state of bOBP takes place at moderate GdnHCl concentration and is followed by the complete unfolding of this protein, whereas conformational perturbations of bOBP induced by low denaturant concentrations are not attributed to the unfolding of the protein globule. In the absence of GdnHCl, the recombinant bOBP is in a stable state with features similar to the native dimeric bOBP. Still, recombinant bOBP in the absence of GdnHCl is characterized by a less ordered secondary structure compared with the wild-type bOBP crystallographic data and a more rigid microenvironment of tryptophan residues. These structural perturbations are responsible for the decreased capability of the recombinant bOBP for dimerization in buffered solutions. We designated this stable recombinant bOBP state in buffered solution as a "trapped" state with incorrect α-helical and β-sheet packing in the protein globule, which may interfere with the formation of the dimeric bOBP native state. The reasons for accumulation of this "trapped" state may lie in a relatively complex domain-swapping mechanism which is required for the monomers to be correctly folded. As a result, in this trapped state bOBP exists as a mixture of monomers and dimers. On the other hand, the intermediate state accumulated at 0.5 M GdnHCl is characterized by the reorganized the bOBP structure, having fewer ordered secondary structure elements, both α-helices and β-strands, compared to the recombinant bOBP both in a buffered solution and in solution containing 1.5 M GdnHCl.

Our analysis revealed that in the presence of low concentrations of PEG-600 (80 mg/mL), shapes of the curves describing the GdnHCl-induced unfolding of the recombinant bOBP were similar to shapes of the corresponding curves recorded in the absence of crowder. However, the half-transition points for the unfolding curves measured in the presence of PEG-600 were shifted towards the higher GdnHCl concentrations ($C_m = 2.4 \pm 0.1$ M, see Fig. 2A; Table 1). Table 2 shows that the values of

**Table 2 Characteristics of intrinsic fluorescence of recombinant bOBP alone and in the different crowding agents.**

| | $\lambda_{max}$, nm ($\lambda_{ex}$ = 297 nm) | Parameter $A$ ($\lambda_{ex}$ = 297 nm) | $r$ ($\lambda_{ex}$ = 297 nm, $\lambda_{em}$ = 365 nm) | $\tau$, nm ($\lambda_{ex}$ = 297 nm, $\lambda_{em}$ = 335 nm) |
|---|---|---|---|---|
| bOBPwt in buffered solution* | 335 | 1.21 | 0.170 | 4.37 ± 0.19 |
| bOBPwt/PEG-600 80 mg/ml | 333 | 1.35 | 0.191 | 4.40 ± 0.17 |
| bOBPwt/PEG-600 150 mg/ml | 332 | 1.40 | 0.195 | 4.09 ± 0.03 |
| bOBPwt/PEG-600 300 mg/ml | 334 | 1.43 | 0.196 | 4.22 ± 0.03 |
| bOBPwt/PEG-4000 80 mg/ml | 334 | 1.29 | 0.194 | 3.68 ± 0.25 |
| bOBPwt/PEG-4000 150 mg/ml | 334 | 1.31 | 0.197 | 3.94 ± 0.10 |
| bOBPwt/PEG-4000 300 mg/ml | 335 | 1.37 | 0.20 | 4.19 ± 0.10 |
| bOBPwt/PEG-12000 80 mg/ml | 335 | 1.28 | 0.192 | 3.96 ± 0.04 |
| bOBPwt/PEG-12000 150 mg/ml | 335 | 1.32 | 0.203 | 4.16 ± 0.07 |
| bOBPwt/PEG-12000 300 mg/ml | 335 | 1.40 | 0.203 | 4.20 ± 0.50 |

**Notes:**
*The data are from *Stepanenko et al. (2014b)*.
The statistical error for fluorescence measurements was assessed and was shown to fall within the range of 0.2–1%. Therefore, the data presented in Table 2 differ significantly.

the parameter $A$ and fluorescence anisotropy $r$ measured for the recombinant bOBP in the presence of 80 mg/mL PEG-600 were somewhat higher than the corresponding values measured in the absence of crowder. The increase in the PEG-600 concentration to 150 mg/mL resulted in the more pronounced increase in the parameter $A$ and fluorescence anisotropy $r$ values. Figure 2B shows that when the 150 mg/mL of PEG-600 are added to the solution of the recombinant bOBP, the pre-transition region of the unfolding curve flattens and the transition happens at higher GdnHCl concentrations than the unfolding in the presence of the 80 mg/mL PEG-600 ($C_m$ = 2.8 ± 0.1 M, Table 1).

Curiously, the curves describing the recombinant bOBP refolding from the completely unfolded state and recorded in the presence of 80 or 150 mg/mL of PEG-600 did not coincide with the quasi-equilibrium unfolding curves recorded under the similar conditions. However, these refolding curves were close to the curves describing unfolding and refolding of the recombinant bOBP alone (i.e., in the absence of crowding agent; Figs. 2A and 2B).

Figure 2C shows that the transition curves describing equilibrium unfolding and refolding of the recombinant bOBP in the presence of 300 mg/mL PEG-600 coincide and have sigmoidal shape. Furthermore, these transitions happened at significantly higher GdnHCl concentrations than transitions recorded in the presence of 80 or 150 mg/mL of this crowder ($C_m$ = 2.9 ± 0.1 M, Table 1; Fig. 6). Table 2 shows that the values of parameter $A$ and fluorescence anisotropy $r$ determined in solutions containing 300 mg/mL PEG-600 were further increased compared to values of these parameters measured at lower PEG concentrations or in the absence of crowding agent. We also observed a slight decrease in the fluorescence lifetime of recombinant bOBP with increasing concentration of PEG-600 from 80–300 mg/mL (Table 1). These data, together with the observed changes in parameter $A$ and fluorescence anisotropy $r$ values, suggested that some compaction of the protein globule took place in the presence of the crowding agent, which resulted in the decrease in a distance between the quenching groups of the protein and its tryptophan residues.

Interestingly, the ANS fluorescence intensity, added to the protein solution in the presence of denaturant and PEG-600 at all concentrations tested, remained substantially unchanged (Fig. 2). These data are likely to reflect the fact that the presence of this crowding agent prevents the possibility of the direct interaction of the molecules of low molecular weight dye ANS and the protein.

## bOBP unfolding in the presence of PEG-4000 and PEG-12000

Addition of the increasing concentrations of PEG-4000 and PEG-12000 was accompanied by the increase in the values of the parameter $A$ and fluorescence anisotropy $r$, as well as the value of fluorescence lifetime (see Table 1). It is worth noting that the value of fluorescence lifetime for recombinant bOBP in the presence of 80 mg/mL of PEG-4000 significantly below the corresponding value for this parameter for bOBP in the presence of 80 mg/mL of PEG-12000, and especially in the presence of 80 mg/mL of PEG-600. However, at elevating the concentration of PEG-4000 and PEG-12000 up to 300 mg/mL the value of the fluorescence lifetime of recombinant bOBP increased to the values typical of the protein in the presence of 300 mg/mL of PEG-600. These data may reflect the different effect of the crowding agents with diverse molecular weights on the structure of the protein.

Curiously, when the unfolding-refolding process of the recombinant bOBP was analyzed in the presence of 80 mg/mL of PEG-4000 or PEG-12000, the corresponding transitions curves coincided with each other and with curve describing the equilibrium unfolding-refolding processes in the recombinant bOBP alone ($C_m$ = 2.3 ± 0.1 M, Table 1; Figs. 3A and 4A). Subsequent increase in concentration of PEG-4000 and PEG-12000 to 150 mg/mL did not change the shape of corresponding curves, but lead to an insignificant and equal for both crowding agents shift of the unfolding transition to higher GdnHCl concentrations and slight flattering of the pre-transition regions (see Figs. 3B and 4B). Furthermore, the half-transition point for the bOBP unfolding in the presence of 150 mg/mL of PEG-4000 or PEG-12000 is observed at a significantly lower GdnHCl concentrations than in the presence of 150 mg/mL of PEG-600 ($C_m$ = 2.6 ± 0.1 M, Table 1; Fig. 6).

The half-transition value evaluated in the presence of maximal studied concentration of PEG-4000 and PEG-12000 can not be determined because of the GdnHCl concentrations needed for the complete unfolding of this protein cannot be not reached due to the high viscosity of solutions. Still, Fig. 3C shows that when PEG-4000 concentration was increased to 300 mg/mL the curve describing the equilibrium unfolding-refolding transitions of bOBP became sigmoidal and coincided with the corresponding curve describing equilibrium unfolding-refolding of this protein in the presence of 300 mg/mL PEG-600 (see also Fig. 6). Although the GdnHCl-induced unfolding curve of the recombinant bOBP in the presence of 300 mg/mL PEG-12000 was also sigmodal (see Fig. 4C), the corresponding transition occurred at significantly lower GdnHCl concentrations (Fig. 6).

Previously we showed that the recombinant bOBP exists as a mixture of monomeric and dimeric forms because of this protein is in a stable native-like state with reduced dimerization capability (Stepanenko et al., 2014b). The compact dimeric state of the

recombinant bOBP is formed under the mild denaturing conditions, namely, in the presence of 1.5 M guanidine hydrochloride (GdnHCl). This process requires bOBP secondary and tertiary structure restructuring and is accompanied by the formation of a stable, more compact, intermediate state that is maximally populated at 0.5 M GdnHCl. In our unfolding-refolding experiment, this state is manifested as a local minimum at 0.5 M GdnHCl on the GdnHCl dependences of the bOBP fluorescent characteristics. It worth noting that the fluorescence anisotropy value measured for bOBP in this native dimeric state (i.e. in the presence of 1.5 M GdnHCl) exceeds that of bOBP in buffered solution in the absence of denaturant. The presence of crowding agents induced the increase of the fluorescence anisotropy values of bOBP even in the absence of GdnHCl. This may reflect that crowding agents are able to shift monomer-dimer equilibrium toward the formation of native dimeric bOBP. Flattering of the unfolding curves of bOBP in the presence of elevated concentrations of crowding agents indicates the unfolding of bOBP follows the two-state mechanism without accumulation of any intermediate states. These observations provide further support for the crowding agent-induced reorganization of bOBP to native dimeric state. This effect depends on the crowding agent used and on its concentration.

In the case of high concentrations of PEG-4000 and PEG-12000, the high values of parameter $A$ and fluorescence anisotropy $r$, as well as the sigmoid shape of the corresponding unfolding curves testify for the fact that crowding agents stimulates preferential transition of the protein to its native dimeric form. As a result, under these conditions, bOBP unfolds according to the all-or-none model (Figs. 4 and 6). However, at lower concentrations of these crowding agents, only slight increase of part of native dimeric state of bOBP occurs.

The ANS fluorescence intensity in the presence of PEG-4000 or PEG-12000 shows almost no dependence on denaturant concentration (Figs. 3 and 4), which is further support for the interruption of any interaction of the ANS molecules and the protein in the presence of studied crowding agent.

Curiously, similar to the results reported in our previous study (*Stepanenko et al., 2016c*), analysis of the recombinant bOBP unfolding in the presence of various concentrations of different crowders revealed that the GdnHCl dependence of various structural characteristics depends on the incubation time of this protein in the presence of the denaturant (see Figs. 2–4). In fact, during the unfolding in crowded milieu, equilibrium and quasi-equilibrium values of the analyzed structural characteristics of the recombinant bOBP were reached after the incubation of this protein in the presence of the desired GdnHCl concentration for 72 hrs. This analysis also revealed the presence of noticeable hysteresis between the curves describing the unfolding and refolding of bOBP when the corresponding measurements were conducted after incubation of the corresponding solution for 1 hour before the measurements (data are not shown).

Figure 5A shows that the tertiary structure of the recombinant bOBP was not affected by low concentrations (80 mg/mL) of PEG-600, PEG-4000, and PEG-12000. However, although the near-UV CD spectra of this protein measured in the presence of high

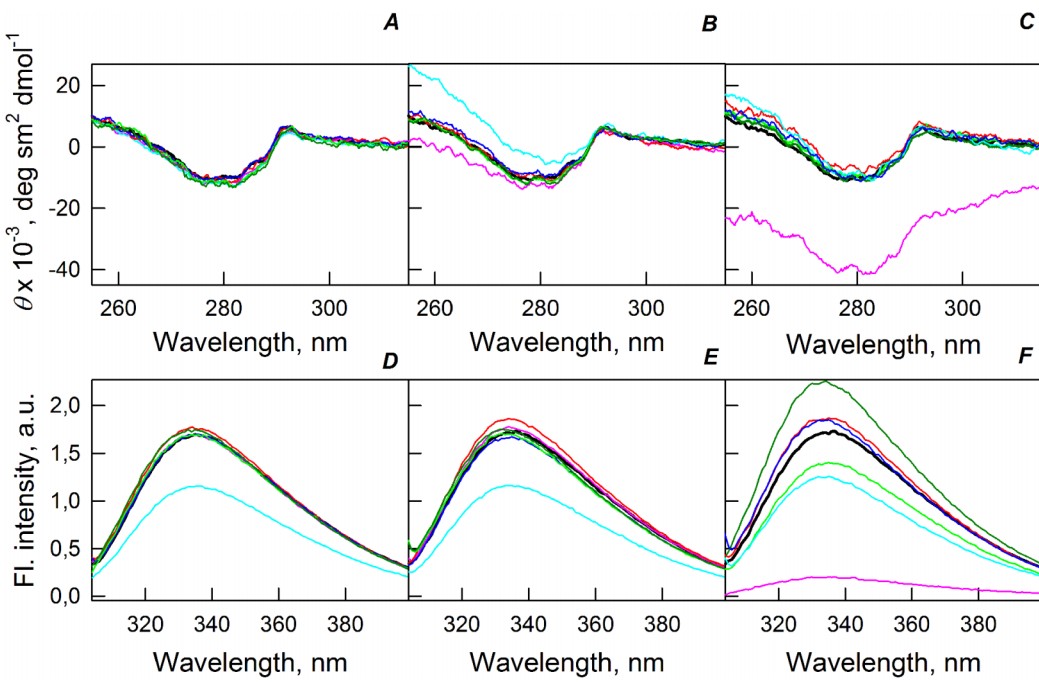

**Figure 5 Changes in the near-UV CD spectra (A–C) and the tryptophan fluorescence spectra (D–F) of bOBP alone (black lines) and in the presence of PEG-600 (green colors), PEG-4000 (red colors) and PEG-12000 (blue colors).** The measurements were preceded by incubating the protein in a solution with crowding agent at 4 °C for 1 h (PEG-600–light-green, PEG-4000–pink, PEG-12000–light blue) and 72–96 h (PEG-600–green, PEG4-000–red, PEG-12000–blue). The concentrations of crowding agents were 80 mg/mL (A), 150 mg/mL (B) and 300 mg/mL (C).

concentrations of crowding agents soon after mixing (~1 h) were different from the corresponding spectrum measured for bOBP alone (see Figs. 5B and 5C), this structural difference disappeared after the prolonged incubation of this protein under the corresponding conditions. The secondary structure of the recombinant bOBP are not changed in the presence of PEG-600, PEG-4000 and PEG-12000, as evidenced by the coincidence of the values of the ellipticity in the far-UV spectrum region recorded for the protein in a buffer solution and in the presence of all crowding agents at all concentrations tested (Figs. 2–4).

The existence of some dependence of the bOBP structure on the time of incubation in the presence of crowders was further supported by the analysis of the intrinsic tryptophan florescence (see bottom panels in Fig. 5). Increase in the incubation time of the recombinant bOBP in the presence of 80 or 150 mg/mL of crowding agents generates fluorescence spectra that practically coincide with the spectrum of intrinsic fluorescence of the protein alone. However, when concentration of the crowding agents was increased to 300 mg/mL, the intensity of the tryptophan fluorescence was noticeably enhanced. In fact, the intensities of the fluorescence spectra measured in the presence of high concentrations of PEG-4000 and PEG-12000 were slightly higher, and spectra measured in the presence of 300 mg/mL PEG-600 markedly exceeded the bOBP fluorescence intensity in the solution without crowding agents.

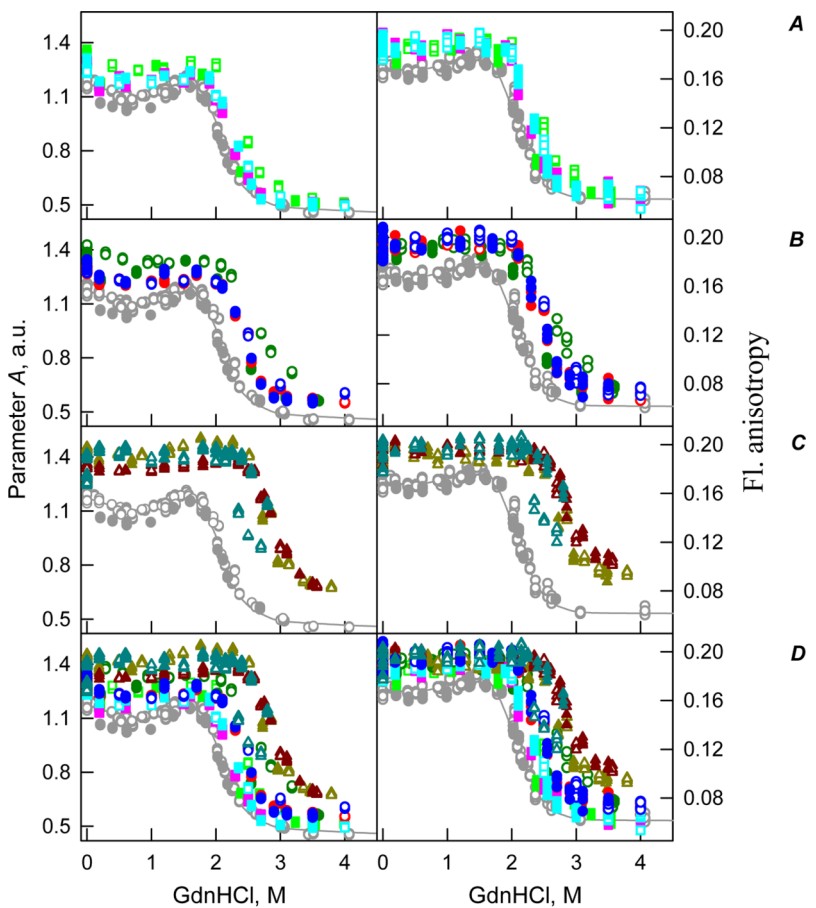

**Figure 6 GdnHCl-induced unfolding–refolding of the recombinant bOBP alone (gray circles; the data are from *Stepanenko et al. (2014b)*) and in the presence of crowding agents PEG-600 (green colors), PEG-4000 (red colors) and PEG-12000 (blue colors).** The protein conformational changes were followed by the changes in parameter *A* and fluorescence anisotropy at the emission wavelength 365 nm ($\lambda_{ex}$ = 297 nm). The measurements were preceded by incubating the protein in a solution with the appropriate GdnHCl concentration at 4 °C for 72–96 h. The open symbols indicate unfolding, whereas the closed symbols represent refolding. Applied concentrations of crowding agents were 80 mg/mL ((A) squares, PEG-600–light green, PEG-4000–pink, PEG-12000–light blue), were 150 mg/mL ((B) circles, PEG-600–green, PEG-4000–red, PEG-12000–blue) and were 300 mg/mL ((C) triangles, PEG-600–dark yellow, PEG-4000–brown, PEG-12000–dark blue). (D) represents all intrinsic fluorescence data for comparison purposes.

## CONCLUSIONS

Our analysis revealed that effects of crowding agents on the structural properties of the recombinant bOBP and on the unfolding-refolding processes of this protein depend on the crowder concentration and size. Being added at low concentrations (80 mg/mL), PEG-600 significantly stabilizes the native sate of the recombinant bOBP judging by the dramatic increase in the corresponding half-transition value. However, at low concentrations, PEG-600 did not influence the mechanism underlying the unfolding-refolding process. This is evidenced by the mismatch of the transition curves describing the bOBP unfolding and refolding. Low concentrations (80 mg/mL) of PEG-4000 and PEG-12000 possess comparable

effects–they do not affect the equilibrium unfolding-refolding pathway but lead to moderate increase in the stability of recombinant bOBP to denaturing effects of GdnHCl. The character of changes of the protein fluorescent parameters such as parameter $A$, fluorescence anisotropy $r$, and fluorescence lifetime reflect different modes of action of different crowding agents analyzed in this study. It is likely that some aspects of the PEG-4000 and PEG-12000 action can be associated with the increased solution viscosity in the presence of these agents, whereas PEG-600 may act through some other mechanisms.

Moderate concentrations (150 mg/mL) of crowding agents lead to further increase in the conformational stability of the recombinant bOBP. Under these conditions, PEG-600 possesses more pronounced stabilizing effects than PEG-4000 and PEG-12000 do. At the highest concentrations of crowding agents analyzed in this study (300 mg/mL), their effects on bOBP were somewhat changed. In fact, our data show that even in the absence of denaturant, there is a substantial compaction of a protein globule and a shift of the conformational equilibrium towards the native dimeric form of the bOBP. Furthermore, the bOBP unfolding curves measured in the presence of high concentrations of crowding agents become sigmoidal, suggesting that the unfolding of this protein under such conditions can be described as an all-or-none transition. Curiously, these changes were essentially dependent on the size of crowding agents, with PEG-12000 possessing smallest stabilizing effects.

Therefore, the effect of crowding agents on the structure and conformational stability of the recombinant bOBP depends on two factors: (i) Size of the crowder, with the smaller crowding agents being more effective in the stabilization of the bOBP native dimeric state; and (ii) on the concentration of the crowding agents, with the higher crowder concentrations typically possessing stronger stabilizing effects.

### Funding
This work was supported by a grant from the Russian Science Foundation RSCF No 14-24-00131. The funder had no role in study design, data collection and analysis, decision to publish, or preparation of the manuscript.

### Grant Disclosures
The following grant information was disclosed by the authors:
Russian Science Foundation RSCF: 14-24-00131.

### Competing Interests
Irina M. Kuznetsova, Vladimir N. Uversky and Konstantin K. Turoverov are Academic Editors for PeerJ.

### Author Contributions
- Olga V. Stepanenko conceived and designed the experiments, performed the experiments, analyzed the data, wrote the paper, prepared figures and/or tables, reviewed drafts of the paper.

- Denis O. Roginskii performed the experiments, analyzed the data, prepared figures and/or tables, reviewed drafts of the paper.
- Olesya V. Stepanenko performed the experiments, analyzed the data, prepared figures and/or tables, reviewed drafts of the paper.
- Irina M. Kuznetsova conceived and designed the experiments, analyzed the data, prepared figures and/or tables, reviewed drafts of the paper.
- Vladimir N. Uversky conceived and designed the experiments, performed the experiments, analyzed the data, wrote the paper, reviewed drafts of the paper.
- Konstantin K. Turoverov conceived and designed the experiments, analyzed the data, reviewed drafts of the paper.

### Data Deposition

All the data generated in this study are reported in figures and table included to the manuscript.

### Supplemental Information

Supplemental information for this article can be found online at http://dx.doi.org/10.7717/peerj.1642#supplemental-information.

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
