# Peer review of "Structure and stability of recombinant bovine odorant-binding protein: III. Peculiarities of the wild type bOBP unfolding in crowded milieu"

_PeerJ, doi:10.7717/peerj.1642_

## Round 0.1 · original submission · Major Revisions

I recommend that the authors revise the article according to the requests of the three reviewers. I agree with the reviewers that the authors need to further analyze their data to make it less descriptive.

Reviewer 1 ·

Basic reporting

The study carried out by Stepanenko and co-workers is well performed. Here, the authors investigated the effect crowded environments -modelled by EG-600, PEG-4000 or PEG-12000- on denaturation of the recombinant variant of bOBP.

Experimental design

The recombinant variant of bOBP has decreased dimerization potential and exists as a mixture of monomeric and dimeric forms in solution. In particular, this occurs bellow 1.6M GdmCl, (e.g. at 0.5M, Figure 6, January 2014 | Volume 9 | Issue 1 | e85169). Indeed, the picture is more complex because there exists certain irreversibility in the refolding process. As this reviewer could understand, the protein refolds to the dimer when folding is performed from protein denatured at moderated denaturant concentrations. Given the intrinsic difficulties of this system, some points should be taking into account by the Authors before publishing this work. Caution should be taken when the reversibility of unfolding/folding is studied. In particular, stability, the difference in free energy of unfolding, is measured only under equilibrium conditions.

Validity of the findings

-¿Is the monomer an out of pathway species? When the protein is refolded, from low or high denaturant concentrations different proportion of dimer and monomer may be obtained.
May be inferred from anisotropy measurements the presence or the absence of the dimer? How can authors be sure that the species under study are the same at the different crowding conditions (the species seems to present similar CD and fluorescence signals). If monomer-dimer ratio cannot be determined, Authors should include a paragraph adding more clarity about this obscure point.

-Authors wrote “unfolding-refolding pathways of the recombinant bOBP and its monomeric forms are similar and do not depend on the oligomeric status of the protein, suggesting that the information on the unfolding-refolding mechanism is encoded in the structure of the bOBP monomers”. However, as this reviewer can understand, there is a stretch of the protein that guides it to the dimeric form. This fact makes that the folding/unfolding mechanism depends, at least in part, of interactions that are not present in the monomeric form. Thus, folding processes of the monomer and dimer seem to be critically different. Please clarify this point.

-Unfolding seems to be very slow in crowding conditions, as judged by the hysteresis between the curves describing unfolding and refolding (very clear when measured only upon 1h incubation). This fact indicates that the process is not under equilibrium. Authors should correct the sentence in lines 282-283: “282 … presence of noticeable hysteresis between the curves describing the equilibrium unfolding and refolding of bOBP when the corresponding measurements were conducted after incubation of the”.

- If it were possible, Authors should include a simple manual-mixing unfolding experiment showing the unfolding traces in different media. This simple experiment may be sufficient to envision this point.

-For experiments in those no evidence of hysteresis was found, Authors should determine the difference in free energy of unfolding and mNU parameter for the reaction.

-¿Has the effect of PEGs on ANS binding (crowder-induced interruption) been observed previously for other proteins or particular conformational states (e. g. molten globules)?

Additional comments

-Results and conclusions should be included in the abstract.
-Delete “…наблюдали незначительное уменьшение величины времени жизни флуоресценции bOBP при повышении концентрации” (lines 232-233), because this paragraph is duplicated.
- When energetic frustration is mapped on bOBP protein an interesting picture is observed. I suggest the Authors to include this information in the manuscript (see http://www.frustratometer.tk/).

Reviewer 2 ·

Basic reporting

-The article is not very well written in English and needs correction. See an example in the paragraph between lines 226 and 238. In addition to English errors, there is the use of the Cyrillic alphabet.

Experimental design

1.In my opinion, from the results it is clear that 1h incubation is not enough to reach equilibrium. That is the logic reason for unfolding and refolding curves not being superimposable. Therefore, the manuscript is better without those data. This solution will avoid confusion and allow for less crowded (and thus cleaner) figures.

2.Most of the curves have few data points in the post-transition region. Could the authors comment on how they deal with that when fitting the data?

Validity of the findings

-Table 1. In my experience, errors in these experiments are of about +/- 5%. Thus, most of the results appear to be the same inside the error. This should be taken in account when writing the Conclusion.

-Figure 6. Have the authors considered the presence of an intermediate inside the pre-transition of the bOBP alone data? How the hypothesis of an intermediate influences the changes (higher signal and flattening) caused by crowding agents in the pre-transition?

·

Basic reporting

The authors have measured by cd and fluorescence spectroscopy the GuaHCl denaturation of recombinant Bovine odorant-binding protein (OBP) in the presence of three concentrations of PEG-600, PEG-4000 and PEG-10K. Their goal was "to shed light on the potential influence of model crowded environment on the unfolding-refolding equilibrium". It is well known that neutral solutes, such as sugars, polyols and other neutral solute (neutral in the sense that they are preferentially excluded from protein surface) preserves enzyme activity an stabilizes proteins against unfolding. Although the effects of cosolvents on protein stability have been quantitatively described considering the colligative properties of cosolvents and, that the increased stability of proteins in such co-solvents have been explained in terms of excluded volume models, this phenomenon is not fully understood yet. Moreover, it is not clear yet the contribution of protein hydration energies, which excluded volume models do not account for. Apparently, the authors are aware of this scenario. Concerning experimental approaches, such studies are carried determining, quantitatively, the effect of co-solvents at varied concentrations on the folding-unfolding equilibrium, and correlating equilibrium constants with co-solvent colligative properties.
The present study fails in these methodological aspects, is just descriptive and built conclusions that are not supported by the data presented.

Experimental design

The experiments of GuaHCl denaturation of OBP were carried out in the presence pegs. However, the data were not analyzed considering established models of protein unfolding, which would allow the determination of the mechanism of folding-unfolding, and the relevant equilibrium constants. Since equilibrium constants were not determined from the experiments, the authors are restricted to make ill defined comparisons like “the half-transition point” “is significant lower than”, when visual inspection of the data does not show that (for instance, see lines 261-264). Without the determination of the equilibrium constants from the data any discussion will be only descriptive, as it is indeed, not allowing any conclusion on the mechanism of solute stabilization, and making hard any comparison amoung effects of different solutes.
The figures are crowded, making difficult to make comparisons amoung the different co-solutes and ar different concentrations. Figure 5 brings few information on the denaturation process in the presence / absence of co-solute. Probably, It would be informative the inclusion of one figure showing a typical experiment of protein denaturation measured by cd in the far-uv.

Validity of the findings

The lack of quantitative data analysis is more compromising when the authors state that the protein may exist in monomers and dimers. For instance, the authors state many times that the measured unfolding-folding curves are sigmoidal although none of the data were analyzed through curve-fitting. In one instance, the ms states that “the equilibrium unfolding-refolding transitions of bOBP became sigmoidal...” (line 266-269) without showing that there was a situation were the curve is not sigmoidal.
Concerning the existence an equilibrium between monomers and dimer, most probably, the shape of the unfolding isotherm will not appear a single sigmoid. Thus, the first conclusion of the ms “ size of the crowder, with the smaller crowding agents being more effective in the stabilization do the bOBP native dimeric states (lines 329-330)” is not supported by the data. Moreover, none of the experiments described in the manuscript allows to distinguish denaturarion of monomers from dimers.

Additional comments

The study lacks quantitative analysis of the data and the experiments are poorly designed to investigate oligomerization effects. Conclusions like "our data show that even in the absence of denaturant, thereis a substancial comaction of a protein globule and a shift towards native dimeric form of bOBP (lines321-323)"; "the bOBP unfolding curves measured in the presence of high concentrations of crowding agents become sigmoidal, sugesting... an all-or-none transition (lines 324-325)", or "the smaller crowding agents being more effective in the stabilization of the bOBP native dimeric state (lines 329-330)", none of them are supported by analysis of the data, or, even by the set of data presented.

---

## Round 0.2 · accepted · Accept

The manuscript is accepted for publication.

Reviewer 1 ·

Basic reporting

No Comments

Experimental design

No Comments

Validity of the findings

No Comments

Additional comments

In general, Authors have replied carefully my questions. After reading the revision that includes corrections, re-organization of figures and incorporation of new tables with important data, I feel that this work has been clearly improved and I recommend it for publication.

Reviewer 2 ·

Basic reporting

No comments

Experimental design

No comments

Validity of the findings

No comments

Additional comments

No comments